# Design and Preliminary Immunogenicity Evaluation of Nipah Virus Glycoprotein G Epitope-Based Peptide Vaccine in Mice

**DOI:** 10.3390/vaccines13040428

**Published:** 2025-04-18

**Authors:** Seungyeon Kim, Rochelle A. Flores, Seo Young Moon, Seung Yun Lee, Bujinlkham Altanzul, Jiwon Baek, Eun Bee Choi, Heeji Lim, Eun Young Jang, Yoo-kyoung Lee, In-Ohk Ouh, Woo H. Kim

**Affiliations:** 1Division of Vaccine Development Coordination, Center for Vaccine Research, National Institute of Infectious Diseases, National Institute of Health, Korea Disease Control and Prevention Agency, Osong, Cheongju 28159, Chungcheongbuk-do, Republic of Korea; hatmddus135@korea.kr (S.K.); msy1477@korea.kr (S.Y.M.); jiwonbb@korea.kr (J.B.); dmsql2274@korea.kr (E.B.C.); dalgi0519@korea.kr (H.L.); sky11kk@korea.kr (E.Y.J.); leeykyoung@korea.kr (Y.-k.L.); 2College of Veterinary Medicine & Institute of Animal Medicine, Gyeongsang National University, Jinju 52828, Gyeongsangnam-do, Republic of Korea; floresrochellea@gmail.com (R.A.F.); seungyun0218@gnu.ac.kr (S.Y.L.); bujinlkham_1221@gnu.ac.kr (B.A.)

**Keywords:** attachment glycoprotein G, epitopes, Nipah virus, peptide vaccine, vaccine

## Abstract

**Background:** The emergence of several paramyxoviruses, including Nipah virus (NiV), makes continued efforts in vaccine development as part of pandemic preparedness efforts necessary. Although NiV is a zoonotic pathogen with high case fatality, there is still no licensed vaccine. **Methods:** Herein, NiV attachment glycoprotein G (NiV-G), which is crucial to host cell receptor binding, was used to develop Nipah epitope-based peptide vaccines. A total of 39 B- and T-cell epitopes of NiV-G were shortlisted for peptide synthesis and evaluation using in silico analysis. **Results:** The in vitro antigenicity evaluation of the peptide candidates showed eight synthesized peptides (G7, stalk-domain epitopes) with relatively high binding to NiV-G antibody-positive serum (A_450nm_: 1.39–3.78). Moreover, nine-mer (9-mer) peptides were found to be less reactive than their longer peptide counterparts (15–30 aa, G7-1, and G7-4), but 9-mer activity was enhanced with cyclization (NPLPFREYK, A_450nm_: 2.66) and C-terminal amidation modification (NPLPFREYK-NH2, A_450nm_: 1.39). Subsequently, in vivo validation in immunized mice revealed the immunogenicity potential of the G7-1 peptide vaccine (30 aa, NENVNEKCKFTLPPLKIHECNISCPNPLPF) to elicit a strong antigen-specific antibody response against their homologous peptide antigen (I.V., A_450nm_: 1.48 ± 0.78; I.M., A_450nm_: 1.66 ± 0.66). However, antibody binding to recombinant NiV-G protein remained low, suggesting limited recognition to the native antigen. **Conclusions:** This study focused on the preliminary screening and validation of peptide vaccines using single formulations with minimal modifications in the peptide candidates. Our findings collectively show the immunogenic potential of the NiV-G stalk-based epitope peptide vaccine as a novel therapeutic for NiV and underscores the need for strategic design, delivery, and formulation optimization to enhance its protective efficacy and translational application.

## 1. Introduction

Globally, the emergence and re-emergence of zoonoses remains a persistent threat to public safety, with significant socioeconomic consequences. Since its discovery in 1998, Nipah virus (NiV) has caused 754 human infections and over 435 human fatalities, with the most recent death reported in Kerala, India, in September 2024 [1,2,3,4]. The virus first emerged in 1998 in Malaysia and subsequently appeared in Singapore and the Philippines in 1999 and 2014, respectively [5,6,7]. While no new cases have been reported in the aforementioned countries since their initial cases, the sporadic and near-annual emergence of NiV since 2001 has been reported in India and Bangladesh, respectively [8]. Designated as a BSL-4 zoonotic pathogen with epidemic potential, NiV is prioritized by the World Health Organization (WHO) for urgent research and development efforts [9]. Depending on strain and outbreak timeline, NiV infection has a case fatality ratio (CFR) of 9–53% for the Malaysian strain and over 70% for the NiV Bangladesh strain; infection typically manifests with severe respiratory and neurological symptoms, and survivors may experience relapsing encephalitis or delayed neurological complications [1,2,10,11,12]. Despite the severity of the disease, effective licensed vaccines and therapeutics remain unavailable, making continued development efforts necessary.

Alongside Hendra virus in the genus *Henipavirus*, NiV is an enveloped RNA (single-stranded negative-sense RNA) virus within the family *Paramyxoviridae* [13]. While human NiV infection has only been geographically reported in South and Southeast Asia, the virus has been identified in other bat species and mammals, with cross-species spillover primarily driven by bats of *Pteropus* spp. [3,6,14,15,16]. Successful experimental infections of NiV have also been reported in chick embryos, pigs, dogs, guinea pigs, hamsters, ferrets, squirrel monkeys, and African green monkeys, as well as horizontal and vertical transmission in cats, highlighting the remarkably broad species tropism of NiV [17,18,19,20,21]. On the other hand, epidemiological outbreak investigations showed that the virus could be transmitted by direct contact with infected animals, as well as with the excretions or secretions of infected animals or the consumption of and exposure to food products (i.e., meat, fruits, sap, or date palm juice) contaminated by the body fluids of infected animals [22,23].

Parallel to other paramyxoviruses, NiV has a genome that encodes for six structural proteins (Nucleoprotein, N; Phosphoprotein, P; Matrix protein, M; fusion glycoprotein, F; attachment glycoprotein, G; and Long protein, L) and three nonstructural proteins, which are gene products of the P gene known as V, W, and C [24,25]. Two distinct membrane-anchored glycoproteins facilitate the successful infection process and entry of NiV into the host cell: attachment glycoproteins (G) and fusion glycoproteins (F) [26]. Specifically, the harmonized function of glycoproteins commences with the binding of glycoprotein G to the host target cell receptors, ephrin-B2 and ephrin-B3, followed by the fusion of the viral and cellular membranes to allow for viral entry, enabled by glycoprotein F [26,27,28]. The cellular receptors ephrin-B2 and ephrin-B3 are highly conserved, with over 95% amino acid identity across mammalian species, and are highly distributed in hosts’ endothelial cells, smooth muscles surrounding the arterioles, neurons, and several regions of the nervous system, which evidences multispecies susceptibility and the pathological features of NiV infection [26,29].

Throughout history, vaccines and immunization have been the cornerstones of global public health and are the most cost-effective measures to combat numerous infectious dis-eases, eradicate life-threatening diseases (i.e., smallpox), and mitigate infectious disease outbreaks by preventing spread, thereby assuring the surest means to defuse pandemic and epidemic risk [30,31]. The availability of genome sequences and various data obtained with omics approaches have revolutionized the understanding of pathogens’ molecular complexity, including a new paradigm for vaccine and drug discovery. To date, candidates for NiV infection include small molecules and antivirals, monoclonal antibodies, and vaccines in various stages of development [32]. The current development of vaccine pipelines for NiV generally includes vaccines targeting surface glycoprotein G and glycoprotein F by using various platforms using viral vectors, protein subunits, virus-like particles, plasmid DNA, and mRNA technology [33]. While there has been progress on these countermeasures, none are approved or licensed for human use. Most of these are still in the preclinical stage, with only four vaccine (recombinant vesicular stomatitis (rVSV)-vectored live attenuated vaccine, clinical trial identifier: NCT05178901; mRNA-1215, NCT05398796; subunit protein HeV-sG-V, NCT04199169; and ChAdOx1-NiV, ISRCTN87634044), one antiviral drug (Ribavirin), and one monoclonal antibody (mAb 102.4, ACTRN12615000395538) candidates reaching clinical trial, underscoring a continued need for the development of effective vaccines and immunotherapeutics as part of pandemic preparedness efforts [32,33,34,35].

Generally, the development of monoclonal antibodies and traditional vaccination strategies using attenuated viruses, their components, and recombinant protein antigens are time-consuming, labor-intensive, expensive, and based on complex processes [36,37]. The ideal characteristics of NiV vaccine candidates include high immunogenicity, safety, thermostability, the ability to elicit rapid protective immunity after a single dose, and cross-protection against Malaysia and Bangladesh strains [33]. Epitope-based peptide vaccines offer an innovative method for vaccine development by reducing production complexity due to their simpler structure and synthesis, since they bypass complex protein expression and folding requirements, allowing for rapid design, ease of production, and scalable, cost-effective manufacturing compared to conventional protein-based platforms [38,39]. Moreover, peptide vaccines are considered inherently safe and are free of infectious risk, as they do not have the risk of reversion to virulence associated with live attenuated or whole-pathogen vaccines [38,40]. On the other hand, challenges to its application include poor in vivo stability and low intrinsic immunogenicity compared to full-length proteins, as they lack the native conformational structure of the epitope in the protein optimal for B-cell recognition [40,41]. As a result, the effectiveness of peptide vaccines often requires careful design, structural modifications, and the inclusion of potent adjuvants or delivery carriers to enhance antigen presentation and immune activation [38,42,43].

The use of in silico analysis to design and demonstrate potential protective immunity of epitope-based peptide vaccines has been reported in various infectious diseases, including SARS-CoV-2, dengue virus, Crimean-Congo hemorrhagic fever virus, Marburg virus, Zika virus, Powassan virus, and human cytomegalovirus [44,45,46,47,48,49,50]. Similarly, several groups have proposed epitope-based peptide vaccines for NiV by using an immunoinformatics approach; however, there is a critical gap due to a lack of in vivo experimental validation [51,52,53]. In the present study, we address this by designing an epitope-based peptide vaccine for NiV. Firstly, a comprehensive in silico analysis of NiV surface attachment glycoprotein G was carried out to identify and select immunodominant B- and T-cell epitopes to design the peptide vaccines, and a preliminary evaluation of the immunogenicity of the potential vaccine candidates was subsequently performed in mouse models.

## 2. Materials and Methods

A schematic representation of the methodology employed in this study is presented in Figure 1.

### 2.1. Protein Sequence Retrieval and in Silico Epitope Selection for Peptide Design

The sequence used in this study was based on the amino acid sequence of the NiV attachment glycoprotein G Bangladesh strain, NiV-G BD (GenBank Accession: OR947676), obtained from a previous report based on a consensus sequence of several full-length NiV-G Bangladesh strain gene sequences retrieved from NCBI [54]. In silico analysis for humoral immunogenicity (B cells) and cellular immunity (T cells) epitopes was performed in a two-step epitope prediction approach by using established tools in the Immune Epitope Database & Tools (IEDB) analysis resource (https://www.iedb.org/). Firstly, linear B-cell epitope prediction was performed by using BepiPred-2.0, a tool trained on epitopes and non-epitope amino acids from a crystal structure using a random forest algorithm, with the default threshold value (0.5) [55]. The predicted B epitopes with more than 9 continuous amino acid residues and high BepiPred scores were input into NetMHCpan 4.1, a tool trained based on binding affinity and mass spectrometry eluted ligand peptide data, for predicting peptides that bind to the MHC molecule [56]. The predicted NiV-G antigenic regions with B- and T-cell immunogenicity overlap or internal 9-mer sequence within the B-cell peptide sequence with high prediction scores were prioritized for subsequent peptide synthesis, antigenicity preliminary screening, and immunogenicity trials. The NiV glycoprotein G structure was predicted using Alphafold2 in ColabFold (https://colab.research.google.com/github/sokrypton/ColabFold/blob/main/AlphaFold2.ipynb) or shown in a resolved NiV-G protein retrieved from Protein Data Bank (PDB) (7TXZ; https://www.rcsb.org/structure/7TXZ), and the structure was viewed in a Mol* 3D viewer (https://www.rcsb.org/3d-view) [57,58]. All the databases and servers used in this study were accessed from 1 January 2024 to 31 March 2024 for the epitope prediction tools and from 9 November 2024 to 6 December 2024 for the protein structures.

### 2.2. General Procedures for Peptide Synthesis

In the present study, all peptides were prepared by using Fmoc SPPS methods and Rink amide or Chloro-trityl resin with an initial loading of 0.61 mmol/g, unless otherwise noted. Additionally, as part of early-phase screening, the most common and cost-effective peptide modifications, backbone cyclization and C-terminal amidation, were selectively applied to shorter peptides to enhance their stability. Fmoc-protected threonine with a benzyl-protected phosphate group (Fmoc-Thr(POBzl)-OH and other Fmoc-protected amino acids were sourced from Novabiochem (Darmstadt, Germany). Initially, the resins were swollen in N,N′-dimethylforamide (DMF) for 45 min before synthesis. Peptide sequence extension was performed by activating the Fmoc-protected amino acid (5.0 eq.) with 1-O-Benzotriazole-N,N,N′,N′-tetramethyl-uronium-hexafluoro-phosphatewith (HBTU; 5.0 eq.) and hydroxybenzotriazole (HOBt; 5.0 eq.) in 2 mL of DMF for 2 min. This solution was introduced into the free amine on the resin with N-diisopropylethylamine (DIPEA; 10.0 eq.), and the coupling reaction proceeded for 1 h under vortex stirring. Thereafter, the resin was washed with DMF; then, Fmoc deprotection was performed by using 20% piperidine in DMF (once for 10 min and then twice for 3 min). The resin underwent an additional wash before the process was repeated for the next amino acid. Once synthesis was complete, the resin was washed with DMF, methanol, dichloromethane, and ether and was then vacuum-dried.

### 2.3. Peptide Cleavage

Linear peptides were extracted from the resin by using 5% triisopropylsilane (TIS) and 5% H_2_O in trifluoroacetic acid (TFA), with an approximate volume of 2 mL of TFA per 100 mg of resin. The cleavage process was carried out for 2 h. Once cleaved, the mixture was combined with cold ether to precipitate the peptide and was subsequently collected by filtration. The filtrate was washed with cold ether, and the peptide was used for the following cyclization step.

### 2.4. Analytical HPLC Conditions

The purity of the peptides was assessed by using two types of HLPC columns and detection at 230 nm. Firstly, in a Phenomenex column (C18, 250 × 4.60 mm, 5 micron, (Phenomenex, CA, USA)), a 30 min linear gradient elution from 10% to 90% aqueous acetonitrile (0.05% trifluoroacetic acid) was performed at a 1.0 mL/minute flow rate. Secondly, the same gradient elution over 30 min was performed in Vydac HPLC columns (C18, 250 × 10 mm, 5 micron, (Avantor, PA, USA)) but at a flow rate of 2.5 mL/min.

### 2.5. Peptide Synthesis with Thioether-Bridged Bond Formation

As mentioned in Section 2.2, peptide synthesis was performed with the Fmoc protection strategy on Rink amide resin (0.61 mmol/g) to create a linear side-chain-protected peptide. After its coupling with the appropriate amino acids, the N-terminal Fmoc group was removed with piperidine (20%) in DMF, and the peptide was bromoacetylated with bis(bromoacetyl) oxide ((BrCH2CO)2O), prepared from 10 eq. of bromoacetic acid and 5 eq. of Dicyclohexylcar-bodiimide (DIC), at room temperature (RT) for 3 h. The peptide (0.061 mmol) was cleaved from the resin and dissolved in a 10 mL of a water/CH3CN mixture (1:1). The pH was adjusted to pH 8–9 with the dropwise addition of triethylamine to facilitate the spontaneous cyclization of the N-bromoacetylated peptide via intramolecular nucleophilic displacement by cysteine thiol. The cyclization process was monitored using HPLC (Agilent Technologies, Santa Clara, CA, USA). After one hour at RT, the solution was acidified with a 30% acetic acid (ACoH) aqueous solution, lyophilized, and then purified with RP-HPLC.

### 2.6. Preliminary Screening of Synthesized Peptides

The complete list of the peptides synthesized is summarized in Table 1. A preliminary screening of the antigenicity of the peptides was carried out with an ELISA using NiV-G-positive serum (IC50:2277.33) inoculated with commercial Nipah G protein (Nipah Virus Glycoprotein G, Human Fc-Tag (HEK293), REC31637, Native Antigen, Oxford, UK). The ELISA was performed as previously described with a slight modification: the microplate was coated with the synthesized peptides (5 µg/mL) in a coating buffer at 4 °C overnight [54]. The control groups used for the assay included a negative control with coating buffer only (NC) and recombinant NiV-G in coating buffer as the positive control (PC). The assay was conducted in triplicate, and the absorbance at 450 nm was measured for each well by using a microplate reader (SpectraMax i3x, Molecular Devices, San Jose, CA, USA).

### 2.7. In Vivo Validation and Evaluation

The candidate vaccine peptides were inoculated either intramuscularly (I.M.) at 100 µg or intravenously (I.V.) at 400 µg into six-week-old, specific pathogen-free (SPF) male BALB/c (AnNCrlOri) mice (Table 2). The peptide vaccine candidates administered I.M. were adjuvanted with Alhydrogel^®^ (100 µg). The experiment was carried out for a total period of 4 weeks. First, the mice were immunized twice at 2-week intervals; then, serum samples were collected 2 weeks later (Section 3.3). Similar immunization was also carried out intramuscularly in the negative control (NC) group with PBS and in the positive control (PC) group with recombinant NiV-G (25 µg; Nipah Virus Glycoprotein G, Human Fc-Tag (HEK293), REC31637, Native Antigen, Oxford, UK). The immunogenicity of the peptide vaccine candidates in mice was evaluated using a previously described ELISA protocol with modification on the capture antigens used [54]. Briefly, microplates were coated overnight at 4 °C with 100 µL/well of 5 µg/mL of either their homologous peptide for the antigen-specific antibody evaluation or with NiV-G recombinant protein (REC31637) for native antigen–antibody recognition evaluation. Thereafter, routine ELISA was performed on the serum samples following washing with PBS-T, blocking, detection with HRP-conjugated goat anti-mouse IgG (50 µL/well (1:50,000) H + L, ThermoFisher Scientific, San Diego, CA, USA) and colorimetric detection using SureBlueTM TMB 1-Peroxidase Substrate (50 µL/well, Seracare Life Sciences, Gaithersburg, MD, USA). The assay was conducted in triplicates and results measured using a microplate reader (SpectraMax i3x, Molecular Devices, San Jose, CA, USA). All mice were obtained from Konebiotech (Seoul, Republic of Korea) and were housed and cared for in strict compliance with the guidelines for the care and use of laboratory animals. The mice were provided ad libitum access to standard mouse feed and water and were monitored daily for clinical signs, including distress and discomfort, ruffled fur, pain, swelling at the injection site, lethargy, a lack of grooming, and mortality post vaccination. Blood samples were collected by trained personnel with minimal restraint via the tail vein following the IACUC protocol. The animal experiment protocol was approved by the Institutional Animal Care and Use Committee (IACUC) of the Korea Centers for Disease Control and Prevention (IACUC number KDCA-IACUC-24-009).

### 2.8. Statistical Analysis

A normality test—the Shapiro–Wilk test—was used to assess the data distribution. Due to the small sample size and one group not passing normality, non-parametric analyses were performed. The data are expressed as means ± standard deviations, and all statistical analyses were performed with Graphpad Prism Software (version 8.0; GraphPad Software Inc., San Diego, CA, USA) by using the Mann–Whitney test between two groups or Kruskal–Wallis analysis followed by post hoc Dunn’s multiple comparison test. A *p*-value < 0.05 represents statistical significance.

## 3. Results

### 3.1. Prediction and Screening of NiV-G B-Cell and T-Cell Epitopes

Ideal vaccine candidates must effectively induce both humoral and cellular responses. In this study, highly antigenic regions of NiV-G were predicted for use in the design of peptide candidate vaccines. The initial analysis of the NiV-G protein sequence using BepiPred predicted a total of 19 linear B-cell epitopes at the 0.5 threshold (Figure 2a, Appendix A). However, each predicted epitope length range varied from a single residue up to 76 residues; thus, a stringent in-house criterion including only fragments of >9 residues along with high residue scores was set to shortlist the epitopes for further analysis. Based on the aforementioned criteria, five B-cell epitopes were deemed eligible and were designated as Group 1 (G1), G7, G11, G14, and G17. The groups extend from regions in the transmembrane (G1), stalk (G7), and head domains (G11, G14, and G17) of the NiV protein. The residue scores and the specific location of the groups in NiV-G were mapped and are shown in Figure 2b and Figure 2c, respectively. The shortlisted B-cell epitopes were then analyzed for T-cell epitopes. Based on the analysis, the HLA Class I immunogenicity scores were the highest in G11, followed by G17, G14, G1, and G7 (Appendix A). Additionally, 9-mer T-cell epitopes for each group were generated, and the highest-scoring ones for each shortlisted epitope are presented in Figure 2d. From this analysis, a total of 12 T-cell epitopes scored over 0.1 in G1 (KVRFENTAS, IKKINEGLL, KKINEGLLD, and VRFENTASD), G7 (NPLPFREYK, LKIHECNIS, PNPLPFREY, PLKIHECNI, and PLPFREYKP), and G14 (SNCPIAECQ, DSNCPIAEC, and NCPIAECQY). On the other hand, the highest-ranking predicted 9-mers in G11 (LNSTYWSGS and VGDPILNST) and G17 (NTVISRPGQ and QSQCPRFNK) had comparatively low scores: 0.070, 0.013, 0.045, and 0.041, respectively. A complete list of the 9-mer peptides with their specific HLA immunogenicity scores is shown in Appendix A.

### 3.2. Selection, Design, and Synthesis of NiV-G Epitope-Based Peptides

Based on the analyses above, a total of 39 NiV-G epitope-based peptides were synthesized from all the groups (Table 1). Priority was given to the epitopes with high scores in both B- and T-cell epitope prediction analyses. The synthesized peptides include the highest predicted B- and T-cell epitope sequences, G11 (VGDPILNSTYWSGS) and G17 (NTVISRPGQSQCPRFNKC), respectively, and the top-ranked 9-mer epitopes, G1 (KVRFENTAS), G7 (NPLPFREYK, LKIHECNIS), and G14 (SNCPIAECQ). Additionally, for better coverage and representation, peptides 15–30 residues in length located within or close to the N and C termini of the predicted antigenic regions of the groups were also synthesized for evaluation (Appendix A). Further, the design and synthesis of the shorter-length peptides were also modified by performing cyclization and amidation for stability.

### 3.3. Antigenicity Evaluation of NiV-G Peptide Vaccine Candidates

All the synthesized peptides were subjected to an ELISA for antigenicity evaluation. Based on the ELISA results, all G1 peptides were generally not antigenic (Figure 3a), whereas some of the peptide candidates from G14 (G14-3 and G14-2), G11 (G11-5), and G17 (G17-NH2 and G17-1) showed antigenicity compared to the NC but at low levels, with an A_450nm_ detection value lower than 0.5 (Figure 3b,d,e). Compared with all the synthesized NiV-G peptide vaccine candidates, only the peptides in G7 showed relatively high antigenicity, with eight of the synthesized peptides from this group (G7-1, G7-1-C1, G7-1-C2, G7-1-N1, G7-4, G7-4-C2, G7-4-9mer-cyc, and G7-4-9mer-NH2) having an A_450nm_ detection value greater than 1 (A_450nm_: 1.39–3.78) (Figure 3c). Interestingly, of all 9-mer peptide vaccine candidates, only G7-4-9mer (NPLPFREYK) had detectable antigenicity, especially when it was synthesized as a cyclic peptide (G7-4-9mer-cyc, A_450nm_: 2.66) and with a C-terminal amidation modification (G7-4-9mer-NH2, A_450nm_: 1.39). Collectively, the results from this evaluation suggest that a 9-mer peptide may not be as antigenic as longer peptides, but some modifications, such as amidation and cyclization, in its design can increase its antigenicity. Moreover, relative to the PC group (NiV-G, A_450nm_: 3.96), the peptide vaccine candidates synthesized from G7-1 (NENVNEKCKFTLPPLKIHECNISCPNPLPF) and G7-4 (KIHECNISCPNPLPFREYKPQTEGVSNLVG) had analogous antigenicity, with A_450nm_ values of 3.91 and 3.78, respectively, and were thus selected as the superior NiV-G epitope peptide vaccine candidates for further immunogenicity validation.

**Figure 3 vaccines-13-00428-f003:**
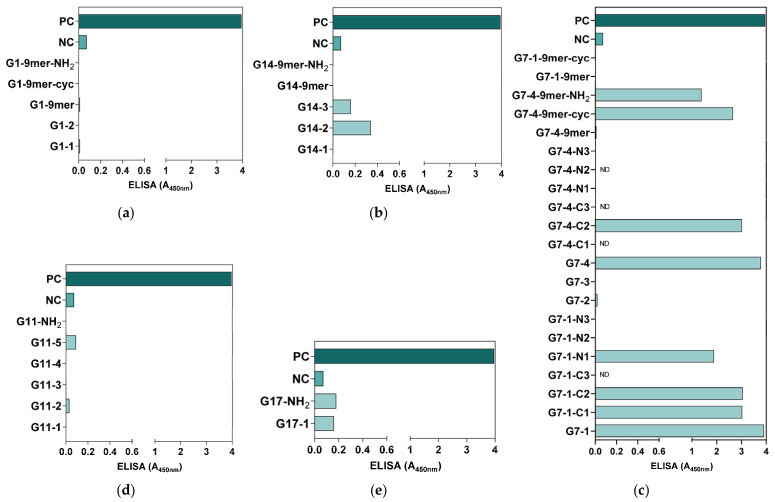
Preliminary antigenicity evaluation of synthesized NiV-G peptide vaccine candidates. The synthesized NiV-G candidate peptide candidates were used as capture antigens in microplates, and a routine ELISA was performed on NiV-G-positive serum. The specific binding of the antibody to the (**a**) G1, (**b**) G14, (**c**) G7, (**d**) G11, and (**e**) G17 synthesized peptides was measured in triplicates, and the data are presented as values of mean absorbance at 450 nm (A_450nm_). ND: Not detected.

### 3.4. NiV-G Peptide Vaccine In Vivo Immunogenicity Evaluation

Based on the antigenicity evaluation, the immunogenicity of the NiV-G epitope peptide vaccine candidates, G7-1 and G7-4, was evaluated in specific pathogen-free (SPF) BALB/c mice. The mice were vaccinated twice with their respective peptide vaccine groups, either I.V. or I.M., and the sera were collected post vaccination for specific antibody titer evaluation with an ELISA (Table 2, Figure 4a). Throughout the in vivo trial, none of the immunized mice showed clinical signs, and no mortality was recorded in all groups. Firstly, the sera of peptide-vaccinated mice were assessed for the evaluation of antigen-specific antibody responses by using an ELISA with their homologous peptide antigens. The mice vaccinated with G7-1 exhibited a robust antigen-specific antibody response following I.V. and I.M. administration with the specific antibody binding values of A_450nm_ of 1.48 ± 0.78 and 1.66 ± 0.66, respectively. In contrast, the mice immunized with the G7-4 peptide showed significantly weak detectable antibodies for either route (A_450nm_: 0.10 ± 0.04 and 0.08 ± 0.1, for I.V. and I.M. respectively) compared to G7-1 (*p*-value < 0.05) (Figure 4b). These findings suggest that the G7-1 peptide vaccine candidate is substantially more immunogenic than G7-4.

Next, the peptide-induced antibodies in the sera were subjected to an ELISA using full-length recombinant NiV-G as the capture antigen to evaluate their ability to recognize the viral protein. Expectably, the result from the assay showed that the PC (NiV-G recombinant)-vaccinated mice exhibited a significantly elevated antibody response (A_450nm_: 3.84 ± 0.03, *p*-value < 0.05), whereas a minimal signal was observed in the NC group (A_450nm_: 0.62 ± 0.09), validating the sensitivity and specificity of the assay. The ELISA results of the peptide-vaccinated groups showed no statistical significance to NC, indicating a low level of recognition and binding to NiV-G viral protein (Figure 4c).

Additional immunogenicity trials using the truncated variants of G7-1 and G7-4 (shorter C and N termini) candidates were also evaluated, but these were less immunogenic and were not cross-reactive to NiV-G (Appendix A). The most immunogenic NiV-G epitope peptide vaccine candidate identified in this study, G7-1, was structurally visualized and mapped onto the cryo-electron microscopy (cryo-EM)-resolved NiV-G ectodomain structure (PDB ID: 7TXZ), revealing its specific localization within the stalk region of the glycoprotein (Figure 5). Overall, the immunogenicity experiment showed that the G7-1 peptide vaccine, an NiV-G stalk-based epitope peptide candidate, is highly immunogenic with regard to eliciting a specific antibody response.

## 4. Discussion

Several paramyxoviruses and novel paramyxoviruses, including NiV, have recently re-emerged and been identified, making a continued effort to develop various strategies to ensure global public health safety necessary [59,60,61,62,63]. Since the discovery of NiV, two genetic lineages with high pathogenicity have been identified, NiV Malaysia (NiV-MY) and Bangladesh (NiV-BD), with NiV-BD causing recent sporadic and lethal outbreaks in Asia, particularly in Bangladesh and India [1,2,3,64]. Immunization continues to be the most economical strategy for preventing and managing various infectious diseases; however, there are no authorized Nipah virus (NiV) vaccines that are currently approved for human use. The availability of NiV genome sequences and their protein structures has paved the way for the ongoing development of vaccine candidates utilizing different strategies and technology with varying levels of efficiency and safety [65,66,67,68,69,70]. However, these approaches may have challenges in terms of scalability, storage stability, consistency in eliciting an immune response, reproducibility, and cost of production, collectively highlighting the need for alternative strategies for vaccine development.

In the present study, we developed an NiV vaccine candidate targeting NiV glycoprotein G, a major antigen for virus–host attachment, by employing an epitope-based peptide vaccine strategy. In contrast to traditional attenuated or recombinant vaccine technology, synthetic peptide-based vaccine technology synthesizes a specific fragment or epitopes of a protein antigen or pathogen with minimal sequences to induce an effective immune response [42,71]. Epitope-based peptide vaccines generally overcome the high cost and complex processes of production and storage stability requirements, including the risk of off-target and adverse effects, as they are synthesized with high precision, high purity, and complete reproducibility [42,71,72]. The flexible nature of peptide vaccine synthesis underscores its advantageous application in disease outbreak scenarios where rapid modifications are necessary to address viral mutations [73].

Similarly to other paramyxoviruses (e.g., parainfluenza, mumps, and respiratory syncytial virus), surface glycoproteins G and F are the main targets in most immunization strategies, as virus-neutralizing antibody responses are largely directed against these glycoproteins [33,69,74,75,76]. Between these two glycoproteins, NiV-G plays a crucial role in host cell receptor attachment; hence, it was the focus of this work [77]. Structurally, it is an oligomeric type II transmembrane glycoprotein characterized by four domains: an N-terminal cytoplasmic tail domain, followed by a transmembrane domain, an extracellular stalk domain, and a C-terminal extracellular globular head domain [13,78]. A previous report on the structure of the NiV-G ectodomain described specific positions of the N-terminal stalk spanning NiV-G amino acid residues 96–147, a neck domain from 148 to 165, a linker region from 166 to 177, and a head domain from 178 to 602 [79].

By using in silico analysis, we were able to streamline peptide design by identifying and shortlisting antigenic regions that are immunodominant in NiV-G. The shortlisted regions in NiV-G predicted to be B- and T-cell epitopes span amino acid positions 7–43 (G1), 139–214 (G7), 300–313 (G11), 371–404 (G14), and 482–499 (G17). By performing the mapping of the location of the groups in a resolved NiV-G protein structure and based on previous reports, we obtained the distribution of the antigenic regions in different domains of the protein. G1 is mainly in the transmembrane domain, whereas G7 extends from the stalk to the head domain, and the remaining groups, G11, G14, and G17, are mainly located in the head domain. Several researchers have reported the immunodominant nature of the NiV-G head domain, including a proof of concept on the vulnerability of this specific domain to neutralizing antibodies in immunized Rhesus macaques [79,80,81].

Here, 39 NiV-G epitope-based peptide vaccine candidates from the above groups with varying residue lengths were synthesized. However, an evaluation with ELISAs showed only eight peptides with high reactivity to NiV-G-positive serum, stressing the importance of experimental validation alongside computational predictions. Interestingly, all the reactive peptides found in this study were from G7, located in the stalk and a part of the head domain region of NiV-G, suggesting that stalk-domain-derived peptides exhibit higher antigen recognition by the antibodies than those from the head domain. This finding could be attributed to the highly conserved nature of the stalk domain compared with the highly variable head domain [82]. Furthermore, shorter-length peptides, such as 9-mers (9 aa), were found to be generally less reactive than longer residue peptides (15–30 aa), but their antigenicity and binding affinity to antibodies were improved when modified through cyclization or amidation. These findings align with previous reports where these modification strategies increased the bioactivity of the peptides [83,84,85]. Structural modifications, such as cyclization and C-terminal amidation, are commonly used in peptide vaccine design to increase the stability and pharmacological activity of the peptide, as cyclization increases resistance to proteolytic degradation and the stability of secondary structures, whereas C-terminal amidation neutralizes terminal charges for enhancing receptor binding and metabolic stability [41,86]. On the other hand, while shorter peptides can offer increased flexibility in their design, longer peptides provide more interaction potential and sustained reactivity, resulting in an effective immune response [87,88].

MHC Class I molecules preferentially bind to 9-mers and MHC-II to longer peptides (12–25 residues), with the 9-mer binding core being preferentially close to the middle of the ligands and the flanking residues extending on both sides of the core [89]. Considering this and the preliminary antigenic evaluation, the longer residue NiV-G peptide candidates in G7, the NiV-G stalk-based peptides, were used to immunize mice for validation. Collectively, the immunogenicity results indicate that the NiV-G peptide candidates, particularly G7–1, a stalk-based NiV-G epitope, were able to elicit an antigen-specific antibody response effectively. Not surprisingly, the level of antibody binding to NiV-G in peptide-immunized mice in this study was markedly lower than those observed in recombinant NiV-G-vaccinated mice. This limited cross-reactivity observed in NiV-G stalk-based peptide-induced antibodies to NiV-G are associated with the lack of a well-defined structure and three-dimensional conformation of the target antigen found in recombinant proteins, in addition to the location of the epitope in the stalk region, which is partially hidden within the protein surface structure [90]. As a result, the antibodies induced by the vaccine candidate exhibit poor binding to the native form of the protein. Additionally, peptides have inherently weak immunogenic properties by themselves compared to protein antigens and have weak membrane permeability, and since they lack conformational integrity, they are more prone to proteolytic degradation, resulting in poor in vivo stability, leading to reduced efficiency in antigen recognition and presentation by antigen-presenting cells [41,91]. Collectively, these intrinsic characteristics of peptides render them inferior to recombinant proteins, unless modifications and optimizations such as the use of carrier molecules to add chemical stability, other delivery systems, or administration with different adjuvants to induce a robust immune response are included in their design and formulation [38,41,92,93].

The head domain is generally the most well reported in targeted vaccine approaches, as it is found to elicit strong neutralizing antibodies and is immunodominant, whereas the stalk domain is considered immunologically subdominant because of its structural positioning near the viral membrane and the partial shielding of the bulky globular head domain [79,94]. However, in recent studies, particularly in those on the design of universal influenza vaccine using Hemagglutinin (HA)-stalk targeted strategy, researchers have highlighted the application and potential of stalk-targeted vaccines in eliciting a robust immune response [82,95,96,97]. In these reports, it was found that a vaccination scheme that can generate immunity to the highly conserved stalk domain could provide broad and durable cross-protective immunity against various influenza strains. Stalk-targeting antibodies were found to inhibit infection by preventing the conformational change necessary for membrane fusion [98]. On the other hand, several structural studies on the stalk domain of various paramyxoviruses have elucidated its functions, including fusion triggering and conformational rearrangement, but there is limited research on its use in vaccine development [99,100]. Taken together, while there have been advancements in the use of the stalk-targeted approach in influenza research, its application in paramyxoviruses is still nascent and remains largely unexplored, especially in in vivo settings.

In this study, the feasibility of an epitope-based peptide for NiV, particularly the potential of a minimally modified NiV-G stalk-derived peptide vaccine as an immunogen, was demonstrated in mice. While this study showed that antigen-specific antibody production was elicited in peptide-vaccinated mice, these antibodies have limited recognition of NiV-G, suggesting suboptimal immunogenicity, underscoring the need for further optimization to enhance protective efficacy. Several limitations inherent to the current study were recognized but fell outside its initial scope. First, the vaccine design utilizes a more targeted approach by synthesizing and employing only predicted B-cell epitopes and HTL epitopes based on their affinity to MHC-I with minimal modification. Second, the peptides were tested as single entities and were not evaluated in combination, i.e., either administered as ‘peptide cocktails’ or synthesized as multi-epitope constructs or with more advanced modifications. Third, only one type of adjuvant (an Alum-type adjuvant, Alhydrogel^®^) was assessed. Going forward, future NiV peptide vaccine development design may benefit from the design of a more robust multi-epitope vaccine, including epitopes with affinity to MHC-II and HTL with cytokine-inducing potential and the use of peptide conjugation in their synthesis to improve immune recognition and response. Despite these limitations, our results provide a valuable starting point for exploring stalk-based peptide vaccine development and underscore the need for a continued effort in the development of effective and broadly protective NiV vaccine candidates.

## 5. Conclusions

In summary, this study demonstrated the design and evaluation of NiV-G epitope peptide vaccine candidates and verified the antigenicity and immunogenicity potential of an NiV-G stalk-derived peptide vaccine in mice. Although the antigen-specific antibodies elicited by the stalk-derived peptide exhibited limited binding to NiV-G protein, our results suggest they hold promise as alternative vaccine candidates following optimization based on multi-epitope design and improved delivery platforms. Additionally, the NiV-G epitope-based peptides demonstrated diagnostic potential, as shown by in-house ELISAs, where they were used as capture antigens, expanding their application beyond vaccination.

## Figures and Tables

**Figure 1 vaccines-13-00428-f001:**
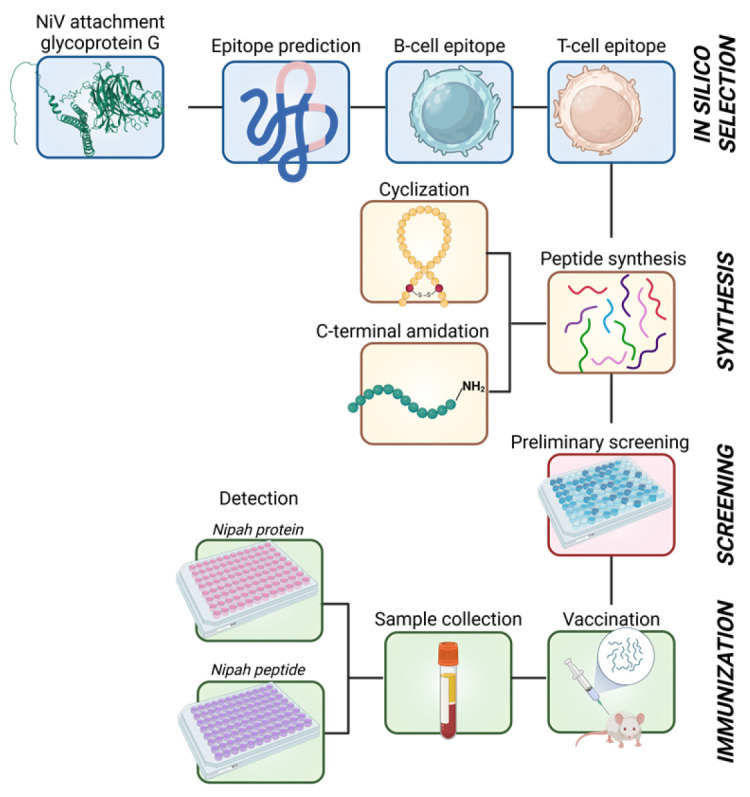
Schematic diagram of the overall methodology employed to design and develop NiV-G epitope-based peptide vaccines. The image was created in Biorender.com with publication rights.

**Figure 2 vaccines-13-00428-f002:**
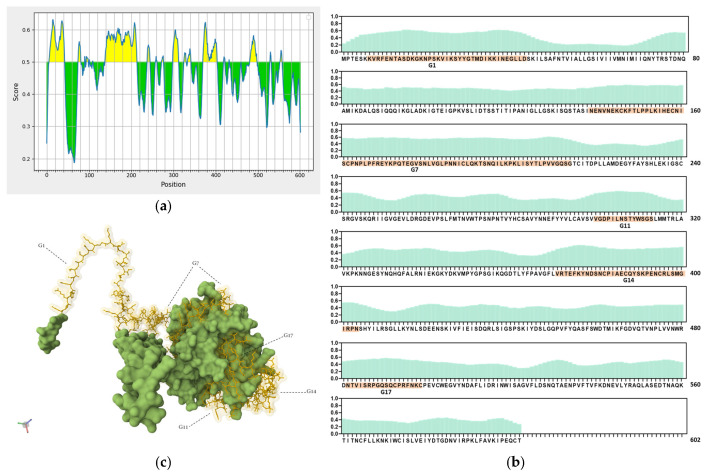
In silico analysis of NiV-G sequence for epitope predictions. (**a**) B-cell epitope prediction score of NiV-G sequence performed by using BepiPred-2.0 at threshold of 0.5. Yellow colors indicate amino acid position over threshold value and indicate high antigenicity. (**b**) Individual BepiPred-2.0 residue scores of NiV-G amino acid sequence. B-cell epitope groups with over > 9 amino acid residues are highlighted in color and marked with their designated groups. Green colors indicate BepiPred residue score. (**c**) Location of B-cell and T-cell epitope groups in predicted NiV-G protein structure in Alphafold. Protein was visualized in Mol 3D viewer and adjusted. Ball and sticks in yellow indicate specific position of residues of groups. (**d**) HLA Class I immunogenicity scores and amino acid sequences of highly ranked 9-mers for each group. G1, Group 1; G7, Group 7; G11, Group 11; G14, Group 14; G17, Group 17.

**Figure 4 vaccines-13-00428-f004:**
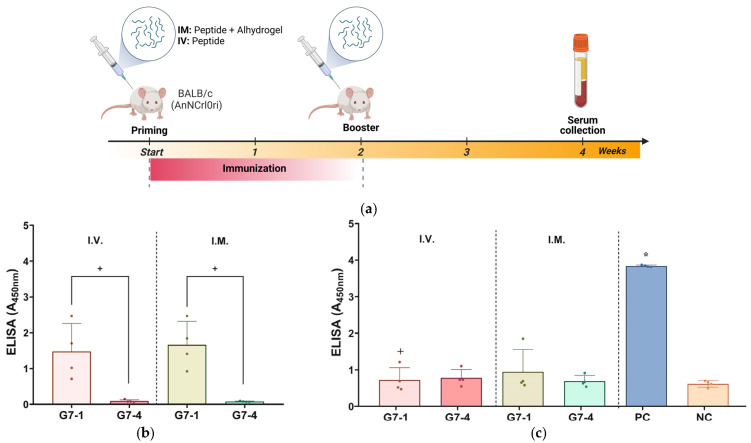
Assessment of immunogenicity of NiV-G peptide vaccine candidates in BALB/c mice. (**a**) Illustration of immunization protocol timeline and sampling schedule for in vivo study in mice. Specific pathogen-free (SPF) mice received peptide vaccinations either intramuscularly (I.M.) with Alhydrogel^®^ adjuvant or intravenously (I.V.) on two occasions, spaced two weeks apart. Individual mouse serum was collected two weeks post booster vaccination to evaluate humoral immune response with ELISA. Image was generated in Biorender with publication license. (**b**) Mouse serum validation against epitope peptide antigens. Individual serum from vaccinated groups was analyzed for specific antibodies against their homologous NiV-G epitope peptide vaccine antigen. Data are presented as values of mean absorbance at 450 (A_450nm_) ± SDs. +, *p* < 0.05 against group. Statistical significance between groups was analyzed by using Mann–Whitney test. (**c**) In vivo immunogenicity evaluation against NiV-G. Individual mouse serum (n = 4) was evaluated for specific antibody binding against NiV-G (REC31637). Results are expressed as values of mean absorbance at 450 (A_450nm_) ± SDs. Statistical significance on all groups was determined by using Kruskal–Wallis test followed by Dunn’s multiple comparison test. *, *p* < 0.05 against negative control (NC); +, *p* < 0.05 against positive control (PC). Each data point represented by a shape within a group in ELISA figures corresponds to serum value from an individual mouse.

**Figure 5 vaccines-13-00428-f005:**
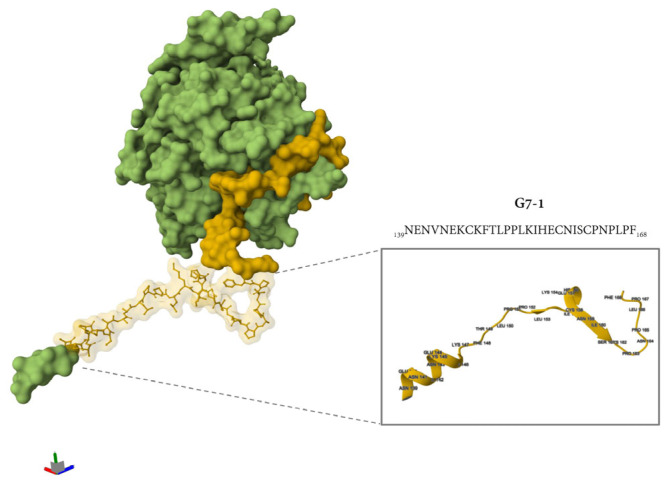
The location of the G7-1 peptide vaccine in the protein structure of NiV-G. The protein structure was retrieved from PDB (7TXZ), and the protein is visualized as its molecular surface representation. Highlighted in yellow is the specific location of G7 groups, and the specific ball and stick representation in yellow is the G7-1 peptide vaccine. The inset shows a simplified representation of protein in a cartoon representation with folding and specific amino acid residues with their specific position number.

**Table 1 vaccines-13-00428-t001:** List of synthesized Nipah virus attachment glycoprotein G (NiV-G) peptides.

Epitope Group	Peptide ID	Sequence	Modification	Amino Acid Position	Peptide Length (aa)
G1	G1-1	KVRFENTASDKGKNPSKVIKSYYGTMDIKK		7–36	30
	G1-1-9mer	KVRFENTAS		7–15	9
	G1-1-9mer-cyc	KVRFENTAS	Cyclization	7–15	9
	G1-1-9mer-NH_2_	KVRFENTAS-NH_2_	Amidation	7–15	9
	G1-2	ASDKGKNPSKVIKSYYGTMDIKKINEGLLD		14–43	30
G7	G7-1	NENVNEKCKFTLPPLKIHECNISCPNPLPF		139–168	30
	G7-1-C1	NENVNEKCKFTLPPLKIHECNISCP		139–163	25
	G7-1-C2	NENVNEKCKFTLPPLKIHEC		139–158	20
	G7-1-C3	NENVNEKCKFTLPPL		139–153	15
	G7-1-N1	EKCKFTLPPLKIHECNISCPNPLPF		144–168	25
	G7-1-N2	TLPPLKIHECNISCPNPLPF		149–168	20
	G7-1-N3	KIHECNISCPNPLPF		154–168	15
	G7-1-9mer	LKIHECNIS		153–161	9
	G7-1-9mer-cyc	LKIHECNIS	Cyclization	153–161	9
	G7-2	REYKPQTEGVSNLVGLPNNICLQKTSNQIL		169–198	30
	G7-3	PNNICLQKTSNQILKPKLISYTLPVVGQSG		185–214	30
	G7-4	KIHECNISCPNPLPFREYKPQTEGVSNLVG		154–183	30
	G7-4-C1	KIHECNISCPNPLPFREYKPQTEGV		154–178	25
	G7-4-C2	KIHECNISCPNPLPFREYKP		154–173	20
	G7-4-C3	KIHECNISCPNPLPF		154–168	15
	G7-4-N1	NISCPNPLPFREYKPQTEGVSNLVG		159–183	25
	G7-4-N2	NPLPFREYKPQTEGVSNLVG		164–183	20
	G7-4-N3	REYKPQTEGVSNLVG		169–183	15
	G7-4–9mer	NPLPFREYK		164–172	9
	G7-4-9mer-cyc	NPLPFREYK	Cyclization	164–172	9
	G7-4-9mer-NH_2_	NPLPFREYK-NH_2_	Amidation	164–172	9
G11	G11-1	NTVYHCSAVYNNEFYYVLCAVSVVGDPILN		277–306	30
	G11-2	STYWSGSLMMTRLAVKPKNNGESYNQHQFA		307–336	30
	G11-3	VPSLFMTNVWTPSNPNTVYHCSAVYNNEFY		262–291	30
	G11-4	YVLCAVSVVGDPILNSTYWSGSLMMTRLAV		292–321	30
	G11-5	KPKNNGESYNQHQFALRNIEKGKYDKVMPY		322–351	30
	G11-NH_2_	VGDPILNSTYWSGS-NH_2_	Amidation	300–313	14
G-14	G14-1	IKQGDTLYFPAVGFLVRTEFKYNDSNCPIA		356–385	30
	G14-2	VRTEFKYNDSNCPIAECQYSKPENCRLSMG		371–400	30
	G14-2-9mer	SNCPIAECQ		380–388	9
	G14-2-9mer-cyc	SNCPIAECQ	Cyclization	380–388	9
	G14-3	ECQYSKPENCRLSMGIRPNSHYILRSGLLK		386–415	30
G-17	G17-1	VVNWRDNTVISRPGQSQCPRFNKCPEVCWE		476–505	30
	G17-NH_2_	NTVISRPGQSQCPRFNKC-NH_2_	Amidation	482–499	18

**Table 2 vaccines-13-00428-t002:** Experimental groups for in vivo validation of Nipah virus glycoprotein G (NiV-G) peptide vaccine candidates.

Group	Peptide Vaccine	Route	Dose	Adjuvant *	Sample Size (n)
G7-1	G7-1 peptide	I.V.	100 µg/dose		4
	G7-1 peptide	I.M.	400 µg/dose	Alhydrogel^®^ (100 µg/dose)	4
G7-4	G7-4 peptide	I.V.	100 µg/dose		4
	G7-4 peptide	I.M.	400 µg/dose	Alhydrogel^®^ (100 µg/dose)	4
PC	NiV-G recombinant protein ^1^	I.M.	25 µg/dose	Alhydrogel^®^ (100 µg/dose)	4
NC	PBS	I.M.			4

I.M., intramuscular; I.V., intravenous; NC, negative control; PBS, phosphate-buffered saline; PC, positive control; * Alhydrogel^®^ adjuvant 2% (Invivogen, San Diego, CA, USA); ^1^ Nipah Virus Glycoprotein G Human Fc-Tag (HEK293), REC31637 (Native Antigen, Oxford, UK).

## Data Availability

All data used in the analyses are provided within this article and its Appendix A. For any additional questions, please reach out to the authors responsible for correspondence.

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
