# Peer review of "Design and Preliminary Immunogenicity Evaluation of Nipah Virus Glycoprotein G Epitope-Based Peptide Vaccine in Mice"

_vaccines, 2025, doi:10.3390/vaccines13040428_

Round 1

Reviewer 1 Report

Comments and Suggestions for Authors

The current manuscript (ID:vaccines-3501823) titled "Design and in vivo immunogenicity evaluation of the Nipah virus glycoprotein G epitope-based peptide vaccine" provides a well-structured and scientifically sound approach to Nipah virus vaccine development, demonstrating the potential of peptide-based vaccines as an alternative strategy. However, there are many weaknesses that should be addressed in the revised manuscript.

Abstract. The abstract does not provide specific numerical data to support its claims. It does not acknowledge any weaknesses of the study.

Introduction. It provides a good overview of the Nipah virus, its impact, and the urgency for vaccine development. The authors need to discuss the difficulties in developing peptide-based vaccines. Some facts (such as virus transmission and outbreaks) are repeated multiple times.

Materials and methods. The study explains each step clearly, from epitope selection to animal testing. But some steps lacks justification. For example, why were certain peptide modifications (cyclization, amination) prioritized over others?  Although IACUC approval is mentioned, more details on animal welfare practices could be provided. 

Results. The visual representation of data makes the results clearer.  Data interpretation is weak in some parts, for example some key results (e.g., why G7 peptides were the most antigenic) are not fully explained.  The study suggests that peptide vaccines have potential but does not provide strong evidence that they are better than existing approaches.

Discussion. This part connects the results to the broader context of vaccine development. But it lacks critical discussion on limitations. While it acknowledges the need for optimization, it does not deeply analyze weaknesses. For example, the relatively low immunogenicity of peptide vaccines compared to recombinant protein vaccines. The possible instability of peptides in vivo. Some claims lack supporting references. For example, the claim that stalk-based vaccines may offer better protection than head-domain vaccines needs more evidence.

Comments on the Quality of English Language

English looks fine to me

Author Response

Manuscript ID: vaccines-3501823 (Response Letter)

We sincerely appreciate and thank you and the editor for your valuable time, and insightful recommendations to improve our current manuscript. The authors have diligently considered the comments, and below are our detailed point-for-point responses and the corresponding revisions/corrections are highlighted in the re-submitted files and in the attached file. We trust these will align with your expectations and contribute to the further improvement of our manuscript.

Reviewer 2 Report

Comments and Suggestions for Authors

The manuscript titled, “Design and in vivo immunogenicity evaluation of the Nipah virus glycoprotein G epitope-based peptide vaccine,” is a description of the authors efforts to design a peptide vaccine from viral glycoprotein sequence in silico, screen these peptides in vitro, then test for immunogenicity in vivo. The effort is a straightforward process that has numerous contemporaries in the literature, which makes the effort someone unique but provides a solid base for the rationale of the vaccine development process. The methods are well described and not particularly objectionable, however, they could be augmented to more thoroughly explore the immunogenicity of the vaccine. The methods used show (Figure 4b) that the antibodies elicited against Nipah virus glycoprotein by the peptides were not distinguishable from the negative control when delivered IM or IV. The authors then run the ELISA for antibody binding to the homologous peptides used for vaccination and, unsurprisingly, find stronger signal. This reviewer is not convinced there is a significant difference between the negative control and the peptides used in the vaccine groups of Figure 4b. The results presented in Figure 4c seem like shooting fish in a barrel (i.e. the test was designed to show better responses). It would be better to focus the analysis on the immune response that would be relevant to infection, which the authors show in Figure 4b would be poor. The authors go on to do a decent job of discussing their scant findings and even go so far as to say, “None of the peptides here were combined….” Given the title of the manuscript and the goal of a vaccine development campaign, I’m surprised the authors did not attempt a combination to achieve some result worthy of publication. As the title suggests, I was thinking that a vaccine was developed but the results do not support such a claim. This manuscript would require significant re-tooling prior to publication. It seems like the obvious paths forward would be to explore more peptide combinations and attempt to find a vaccine that elicits an immune response greater than the negative control, or to focus the effort on the rapid development the authors mention in the discussion, which would require a new title. Please address the lack of data in support of the title of this manuscript prior to publication.

Line 27 and general comment on the Abstract: The authors say, “…but lower compared to recombinant proteins.” This statement has no reference point for the reader. There is no telling what recombinant proteins to which the authors refer or what is lower. Additionally, the abstract should be a description of the entire paper, which the authors do not include; for example, the authors mention immunogenicity studies but nothing of their results. It would not be necessary to recount the results but this would be the opportunity to say that the mice had B or T, or both, cell responses. Please address the confusing references in the abstract and include language to summarize the manuscript.

Author Response

(The authors gave the same response as above.)

Reviewer 3 Report

Comments and Suggestions for Authors

The manuscript describes peptides corresponding to one strain of Nipah virus glycoprotein G as "innovative method" of vaccine production. But the approach is well known. Currently, peptide vaccines against most viral infections including COVID-19 remain unavalable. Peptides could model linear epitopes only but do not correspond to conformational antigenic determinants of full-lengh antigens. Usually,  the peptides are weak antigens and immunogenes. Unfortunately, disadvantages of peptides as vaccine candidates were not described in "Introduction" and were not discussed at all in the text. 

General comments. 

  1. The selection of peptides was based on the only strain of RNA-containing paramyxovirus Nipah but not on the multiple alignment of amino acid residues of different strains and isolated from various regions. So heterologous protection is under question.
  2. Nucleotide sequence and deduced amino acid sequence (GenBank Accession number OR947676) has not been released in open access yet. 
  3. Selection criteria of the peptides studied were not described. How the correspong software works? Surface localization, hydrophilic/hydrophobic profiles, polar and charged amino acids or homology with known epitopes localized by means of epitope mapping with monoclonal antibodies?
  4. Vaccine development depends on infection. risks. But the Nipah virus infection rate and recent dynamics are not descibed.   The only fatality rate was mentioned in "Introduction". The epidemiological data should be carefully checked, 
  5. In the absence of Nipah virion structure it's hardly possible to understand the localization of  "stalk" region of the glycoprotein G  in virions. Is it the transmebrane region? The globular "head" is localized on the surface of envelope, isn't it? So, antibodies are supposed to recognize epitopes in "head" but not in the mebrane-associated "stalk". 

Minor comments are in the attached file. 

Monoclonal antibodies (line 97) can be considered as passive but not active immunization. Antibodies are not common vaccines but rather their consequences. 

Comments on the Quality of English Language

English language and especially scientific terms and abbreviations should be carefully checked. 

Author Response

(The authors gave the same response as above.)

Reviewer 4 Report

Comments and Suggestions for Authors

The results of the study may be useful in the development vaccine candidates against Nipah virus. Data on the search for epitopes and their antigenic properties are interesting.

However, the interpretation of the immunogenicity experiment is misleading and requires correct presentation. Some of the study's сonclusions are not supported by the data.

Major points:

-Lines 322-325.

“However, all peptide vaccinated groups regardless of route…were generally higher to NC group…”. This thesis is invalid since there is no statistical significance between the negative control and the groups immunized with peptides.

-Lines 325-327. “Regardless, G7-1 peptide vaccine administered I.M. have increased detectable specific NIV-G antibody by 51.6% relative to NC group (Fig. 4B)”.

This thesis is also invalid for the reason stated in the previous paragraph.

-Line 426. “…were able to elicit immune response…”

This statement is not supported by the data from the immunogenicity experiment.

-Lines 448-451. These conclusions should be adjusted in the context of the correct interpretation of the immunogenicity data.

-In section 2.7 the dosage/quantity of Alhydrogel used for immunization should be indicated. And also indicate in which cases the adjuvant was used and in which it was not. Adjustments should be made to the Table 2 and Figure 4 in accordance with this information.

-When describing statistics, it is more correct to use SD rather than the standard error of the mean.

Minor points:

-Section 2.7 should specify the method of blood collection.

-It is better to write absorption at 450 nm instead of optical density.

-Check through the text of the MS that the terms in vitro, in vivo, in silico, etc. are italicized. I noticed a number of sentences where this was not the case, lines: 109,111,125,319,393,429,449.

Author Response

(The authors gave the same response as above.)

Round 2

Reviewer 1 Report

Comments and Suggestions for Authors

The authors have responded to my comments in the revised manusc

Author Response

Manuscript ID: vaccines-3501823 (Response Letter)

Response to Reviewer 1 Comments and Suggestions for Authors: Round 2

Thank you for your thorough evaluation and constructive feedback during the revision process. We’re pleased to hear that all concerns have been addressed. Your insights have contributed positively to the clarity and quality of the manuscript.

Reviewer 4 Report

Comments and Suggestions for Authors

Unfortunately, I still cannot agree with the authors in interpreting a substantial part of the immunogenicity data. If there are no significant differences with the negative control, it is not correct to discuss the “numerically higher values” of the experimental groups. It is impossible to discuss the presence of low but detectable levels of cross-reactive antibodies capable of binding to NiV-G, unless significant differences with negative controls are identified. The absence of considerable differences between the groups immunized with peptides with negative control is also clearly seen from the distribution of data in Figure 4C.

With regard to the above the following text fragments should be deleted:

-Lines 29-30, “and a low but detectable level of cross-reactive antibodies to NiV-G when administered I.M. with Alhydrogel® (A450nm: 0.94 ±0.61)

-Lines 361-369, “but all groups exhibited numerically higher values, indicating the presence of low but detectable levels  of cross-reactive antibodies capable of binding to NiV-G. Notably, among the peptide-vaccinated groups, G7-1 administered via the I.M. route displayed the highest binding to  NiV-G (A450nm: 0.94 ±0.61), corresponding to a 51.6% increase relative to the NC group, suggesting enhanced immunogenicity compared with other peptide-vaccinated groups. By comparison, other peptide groups, including G7-1 administered I.V. (A450nm: 0.72±0.34) and G7-4 administered both I.V. (A450nm: 0.78±0.23) and I.M. (A450nm: 0.69±0.16), have only  marginal increases, reinforcing the relative dominance of the G7-1 peptide vaccine when 369 administered I.M. with Alhydrogel® (Fig. 4C)”.

-Lines 379-380, “and that these antibodies have modest cross-reactivity to NiV-G mainly when administered I.M.”

-Lines 478-479, “and induce low but measurable cross-reactive antibodies  against NiV-G most notably following I.M. administration with Alhydrogel®”

-Lines 483-484, “This partial recognition

-Lines 539-540, “the antigen-specific antibodies produced were robust

In section 2.7 the ELISA modifications should be discussed in more detail.

Author Response

Manuscript ID: vaccines-3501823 (Response Letter)

We appreciate the reviewer's critical and constructive feedback on the interpretation of immunogenicity evaluation to improve our manuscript. We agree that it is inappropriate to emphasize 'numerically higher' values as evidence of antibody response in the absence of statistical significance compared to the negative control group. In line with the suggestions, some sections in the manuscript were removed but some areas were rephrased and adjusted to make a more accurate interpretation of our results. We hope this revision provides more clarity while retaining the value of the experimental observations in this early-phase study. The revisions are highlighted in blue text in the revised manuscript and our point-by-point response on the comments is in the attached file.

Round 3

Reviewer 4 Report

Comments and Suggestions for Authors

I understand the authors' desire to somehow interpret the data obtained on the cross-reactivity of peptides with NIV-G. Nevertheless, at the moment, also possibly due to the insufficient data (small group size, multiple comparisons), the presence of cross-reactivity has not been shown.  It is not correct to use the concept of trend in application to insignificant results. This is quite actively discussed now in the scientific community, including in various fields of biological sciences. As one example: doi: 10.1001/jamaoncol.2018.4524.

The phrase (lines 376-381):

“The ELISA results of the peptide-vaccinated groups showed increasing trend (G7-4 IM with Alhydrogel®: A450nm: 0.69±0.16, G7-4 IV: A450nm: 0.78±0.23, G7-1 IV: A450nm: 0.72±0.34, G7-1 IM with Alhydrogel®: A450nm: 0.94 ±0.61). However, none of these vaccinated groups reached statistical significance to NC, indicating a low level of recognition and binding to NiV-G viral protein (Fig. 4C)” 

should be replaced by

“The ELISA results of the peptide-vaccinated groups showed no statistical significance to NC, indicating a low level of recognition and binding with NiV-G viral protein (Fig. 4C)”

Please leave all possible speculation to the readers.

If you really want to give numerical values, add tables with mean and standard deviations for all ELISA experiments in the Supplementary file. However, do not interpret insignificant results as a trend, partial recognition, etc.

Author Response

Manuscript ID: vaccines-3501823 (Response Letter)

We value the Editor and the reviewer for their guidance in presenting our data clearly and cautiously. We recognize and appreciate the feedback, particularly on the interpretations regarding using trends on non-significant findings. We have highlighted the necessary revisions in green text in the revised manuscript and trust these changes address the concern and contribute to a more transparent and accurate presentation of our findings.

Point-by-point response to Reviewer 4 Comments and Suggestions for Authors: Round 2

I understand the authors' desire to somehow interpret the data obtained on the cross-reactivity of peptides with NIV-G. Nevertheless, at the moment, also possibly due to the insufficient data (small group size, multiple comparisons), the presence of cross-reactivity has not been shown. It is not correct to use the concept of trend in application to insignificant results. This is quite actively discussed now in the scientific community, including in various fields of biological sciences. As one example: doi: 10.1001/jamaoncol.2018.4524.

The phrase (lines 376-381):

“The ELISA results of the peptide-vaccinated groups showed increasing trend (G7-4 IM with Alhydrogel®: A450nm: 0.69±0.16, G7-4 IV: A450nm: 0.78±0.23, G7-1 IV: A450nm: 0.72±0.34, G7-1 IM with Alhydrogel®: A450nm: 0.94 ±0.61). However, none of these vaccinated groups reached statistical significance to NC, indicating a low level of recognition and binding to NiV-G viral protein (Fig. 4C)”

should be replaced by

“The ELISA results of the peptide-vaccinated groups showed no statistical significance to NC,indicating a low level of recognition and binding with NiV-G viral protein (Fig. 4C)”

Please leave all possible speculation to the readers.

If you really want to give numerical values, add tables with mean and standard deviations for all ELISA experiments in the Supplementary file. However, do not interpret insignificant results as atrend, partial recognition, etc.

Response: We have removed the phrase in Lines 376-381 and adapted the suggested phrase in Lines 376-378:

"The ELISA results from the peptide-vaccinated groups showed no statistical significance compared to the negative control, indicating a low level of recognition and binding with NiV-G viral protein (Fig. 4C)."